# Physiochemical Responses and Ecological Adaptations of Peach to Low-Temperature Stress: Assessing the Cold Resistance of Local Peach Varieties from Gansu, China

**DOI:** 10.3390/plants12244183

**Published:** 2023-12-16

**Authors:** Ruxuan Niu, Xiumei Zhao, Chenbing Wang, Falin Wang

**Affiliations:** Institute of Fruit and Floriculture Research, Gansu Academy of Agricultural Sciences, Lanzhou 730070, China; rxniu@gsagr.cn (R.N.); zhaoxiumei5@gsagr.ac.cn (X.Z.); wangcb@gsagr.ac.cn (C.W.)

**Keywords:** cold resistance, peach, anatomical structure, physiochemical responses, ecological adaptations

## Abstract

In recent years, extreme weather events have become increasingly frequent, and low winter temperatures have had a significant impact on peach cultivation. The selection of cold-resistant peach varieties is an effective solution to mitigate freezing damage. To comprehensively and accurately evaluate the cold resistance of peaches and screen for high cold resistance among Gansu local resources, nine different types of peach were selected as test resources to assess physiological, biochemical, and anatomical indices. Subsequently, 28 peach germplasms were evaluated using relevant indices. The semi-lethal temperature (LT50) was calculated by fitting the change curve of the electrolyte leakage index (ELI) with the Logistic equation; this can be used as an important index for identifying and evaluating the cold resistance of peach trees. The LT50 values ranged from −28.22 °C to −17.22 °C among the 28 tested resources; Dingjiaba Liguang Tao exhibited the lowest LT50 value at −28.22 °C, indicating its high level of cold resistance. The LT50 was positively correlated with the ELI and malondialdehyde (MDA) content with correlation coefficients of 0.894 and 0.863, respectively, while it was negatively correlated with the soluble sugar (SS), soluble protein (SP), and free proline (Pro) contents with correlation coefficients of −0.894, −0.721, and −0.863, respectively. The thicknesses of the xylem, cork layer, cork layer ratio (CLR) and thickness/cortex thickness (X/C) showed negative correlations (−0.694, −0.741, −0.822, −0.814, respectively). Finally, the membership function method was used to evaluate cold resistance based on the ELI, MDA, Pro, SP, SS, CLR, and xylem thickness/cortex thickness (X/C) indices. The average membership degree among all tested resources ranged from 0.17 to 0.61. Dingjiaba Liguang Tao emerged prominently in terms of high-cold-resistance (HR) membership value (0.61).

## 1. Introduction

The detrimental impact of freezing injury is a primary cause of crop loss. Indeed, low temperature (LT) is acknowledged as a primary factor constraining the geographical distribution of plants [1]. Fruit trees across the globe, especially those in temperate zones, are particularly susceptible. They often suffer varying degrees of LT-induced damage, leading to substantial production setbacks [2]. The fatal consequences of ice formation within intracellular spaces have been extensively documented [3,4]. The screening of cold-resistant rootstocks or varieties is considered one of the most effective strategies for addressing freezing damage [2,5]. Therefore, it is particularly crucial to evaluate and screen the cold resistance of peach resources.

To counteract freezing injury, plants exhibit myriad adaptive responses during cold acclimation. The modifications encompass alterations in membrane composition, augmentation of osmoregulation, regulation of plant hormones, and structural reorganization during prolonged acclimation [5,6,7,8]. After undergoing low-temperature stress treatment, the plant cell membrane undergoes a phase transition, resulting in an elevation of membrane permeability, exosmosis of electrolytes, and enhanced electrical conductivity [9]. The conductivity method is a conventional approach for determining the freezability of in vitro tissues. By fitting the inflection point using the Logistic equation, the LT50 can be obtained to reflect the quantitative relationship between temperature, moisture, and cold resistance [10,11]. This method demonstrates excellent fit, simplicity, and accuracy when applied to various fruit trees [11,12,13,14] and enables an accurate assessment of their cold resistance. The cold resistance of peach may be closely associated with multiple parameters. For instance, MDA, the end product of membrane lipid peroxidation, serves as a marker for assessing cell membrane damage in cold-resistance evaluations [15,16]. Under LT stress, plants accumulate various osmoregulatory substances to enhance their osmoregulation capacity and improve cold resistance [17]. Osmoregulatory factors such as proline, soluble sugars, and soluble proteins play pivotal roles in enhancing cold resistance [14,18]. Simultaneously, trees mitigate cellular damage and maintain normal physiological functions by enhancing antioxidant enzyme activity to scavenge reactive oxygen species (ROS), thereby exhibiting a protective stress response [19]. Antioxidant enzymes, such as superoxide dismutase (SOD), peroxidase (POD), and catalase (CAT), constitute the primary active oxygen scavenging system in plants [20,21]. The activity levels of these enzymes also serve as an indicator for evaluating the cold resistance of fruit trees [22]. Furthermore, plants with strong cold resistance possess corresponding structures to adapt to LT environments. When exposed to prolonged LT stress, plants can mitigate LT injury by modifying their structural characteristics and enhancing their adaptability to the cold environment [23]. Therefore, morphological and anatomical structural characteristics are considered important reference indices for studying cold resistance [24]. However, the inherent complexity of plant cold resistance, being a quantitative trait shaped by various factors, requires comprehensive assessment. Relying solely on a singular index might introduce inaccuracies [25,26].

Peach (*Prunus persica* L.) is a representative stone fruit that is full of nutrients and highly accepted worldwide [27,28]. Based on Food and Agriculture Organization of the United Nations (FAO) statistics, the global area for peach planting was 2.27 million ha in 2020, with peach production of 24.17 million tonnes (http://www.fao.org/faostat/en/#data/QC accessed on 30 November 2021). China is the leading country for peach production, accounting for approximately 34.33% of the world’s total harvest area and 62.1% of the total production (http://www.fao.org/faostat/en/#data/QC accessed on 30 November 2021). However, it frequently suffers from freezing injury due to a lack of cold hardiness during winter. In recent years, the frequent occurrence of LT disasters has seriously affected the development of the Gansu peach industry [29]. China, posited as one of the origin centers of the peach, showcases a remarkable diversity of this fruit [30,31]. The Hexi Corridor region, Gansu Province, located in western China, experiences a long and harsh winter with prolonged soil freezing. The average absolute minimum temperature reaches approximately −30 °C, with occasional lows reaching −35 °C [32]; the region’s peach trees have endured over 2000 years of natural selection and cultivation. Long-term acclimation has bred a number of special local peach cultivation types with strong cold resistance, which are valuable germplasm materials for resistance research and breeding. Hence, the gauging of their cold hardiness and comprehension of their LT adaptive mechanisms are pivotal to unraveling the evolutionary strategies peaches employ in colder climes.

In our research, to obtain an overall view of the metabolic changes related to cold hardiness and evaluation of cold resistance of local Gansu peach resources, we analyzed changes in the physiological and biochemical indices of nine peaches under different degrees of LT stress and the relationship between the LT50 and annual branch structure. Utilizing the average membership function, we also assessed the cold resilience of 28 distinct peach resources.

## 2. Material and Methods

### 2.1. Plant Materials

The germplasm branches (*n* = 28) were sampled from the Gansu Academy of Agricultural Sciences Peach Germplasm Repository, located in Gansu province, China. All the resources’ rootstocks consisted of Hong Hua Shan Tao (*P. davidiana* Franch). On 5th January 2019, a total of 80 branches of equal length (25–30 cm) and diameter (0.8–1.0 cm) were collected from all sides of seven-year-old trees. The branches were cleaned using distilled water and their shoot scissors were sealed with paraffin wax. Each branch was divided into eight groups and each group was carefully wrapped with gauze and plastic bags for protection during subsequent treatments.

The groups were randomly assigned to one of the eight programmable temperature incubators set at temperatures of −5 °C (control), −10 °C, −15 °C, −20 °C, −25 °C, −30 °C, −35 °C, and −40 °C, respectively. Cold treatment lasted for 12 h with cooling or heating rates set at 4 °C/h. After cold exposure, the middle parts of the branches (excluding flower and leaf buds) were sampled promptly and frozen in liquid nitrogen before being stored at −80 °C until further use (electrolyte leakage measurements excluded freezing effects). Branch materials were obtained from three different trees as independent biological replicates.

The experimental materials for physiological, biochemical, and ecological index screening consisted of 9 peach resources selected from different regions and types (Table 1). An expanded assessment was conducted using a total of 28 peach resources, including a rootstock, 7 domesticated peach resources (Table 2), and 20 local resources from the Hexi Corridor in Gansu Province (Table 3; Figure 1).

### 2.2. Physiochemical Analyses

The level of lipid peroxidation, an indicator of oxidative stress, was determined by measuring the malondialdehyde (MDA) content using the thiobarbituric acid (TBA) reaction [33]. Proline (Pro) concentrations, which signal osmotic adjustments, were assessed using the sulfosalicylic acid–acid ninhydrin method [34]. Soluble protein concentrations were evaluated using the Coomassie blue staining method [35], while soluble sugar levels, indicative of carbohydrate metabolism, were ascertained using anthrone colorimetry [36]. The Electrolyte Leakage Index (ELI), a proxy for membrane integrity, was computed as previously detailed [37]. Enzymatic assays for peroxidase (POD), superoxide dismutase (SOD), and catalase (CAT) were conducted using kits provided by Suzhou Ke Ming Biotechnology Co., Ltd., Suzhou, China [38].

### 2.3. Anatomical Structure Analyses

For microscopic examination, five branches from each peach variety were set aside. A 1.0 cm section from the midpoint of every branch underwent treatment with a softening solution (comprising equal parts ethanol and glycerol) until they attained the required flexibility for slicing [39,40]. Tissue staining was conducted using saffron solid green. For each variety, measurements were taken from five branches, using three observation fields per branch, leading to a total of 15 data points per variety. The Digimi Zer 4.5.1 software was employed to measure various anatomical features, including diameter, xylem, cork layer, phloem, and cortical tissue thickness. Subsequent calculations included:CR (Cortex ratio of branches) = (thickness of cortex/branch radius) × 100%;
CLR (Cork layer ratio) = (cork layer thickness/branch radius) × 100%;
XR (Xylem ratio) = (xylem thickness/branch radius) × 100%;
X/C = xylem thickness/cortex thickness.

### 2.4. Statistical Analysis

Data compilation and preliminary statistical analyses were executed using EXCEL 2010 and SPSS Statistics 24.0 software packages. For each experiment, three independent biological replicates were employed. Mean values were discerned and statistically differentiated using Duncan’s Multiple Range Test; a *p*-value of less than 0.05 was deemed statistically significant. All data points are elucidated as the mean ± standard error. Cluster analysis was carried out according to the comprehensive scores of principal component evaluation using Ward’s method. By employing the principles laid out by Zhang et al. [25], subordinate function values were computed to assess the cold resilience of various peach resources using the following equations:Membership function = (X_i_ − X_min_)/(X_max_ − X_min_) (1)
where X_i_ signifies the determined index value, while X_min_ and X_max_ represent the smallest and largest values for a specific index across all evaluated samples. In cases where an index is inversely proportional to cold hardiness, the anti-membership function is utilized:Anti-membership function = 1 − [(X_i_ − X_min_)/(X_max_ − X_min_)] (2)

## 3. Results

### 3.1. Changes in Physiological and Biochemical Indices at Different LTs

For the nine resources studied, the Electrolyte Leakage Index (ELI) under LT stress remained relatively stable until a significant surge was observed as temperatures dropped to −25 °C. This increase reached its peak at −35 °C. Notably, when temperatures further decreased to −40 °C, the ELI values for most resources showed only slight differences compared to those at −35 °C (Figure 2A). In quantitative terms, the average ELI at −35 °C rose by 121.18% compared to its value at −5 °C.

Similarly, the MDA content in most resources saw a gradual rise until temperatures reached −20 °C. At this point, the MDA content increased by 82.31% compared to the levels seen at −5 °C, peaking at −35 °C. However, when temperatures hit −25 °C and −30 °C, there was a notable reduction in MDA levels compared to those at −20 °C (Figure 2B).

Parallel to this, as temperatures decreased, both soluble sugar and protein concentrations generally rose, with a decline observed at −30 °C. The peak levels of soluble sugar and protein were found in samples subjected to −25 °C, while the lowest levels were recorded at −5 °C and −35 °C, respectively. The proline concentration exhibited a distinctive pattern: an initial increase, followed by a subsequent decrease, and then another rise. Temperatures below −5 °C significantly stimulated the accumulation of proline. A notable increase in proline content was observed when the temperature dropped below −15 °C, which was subsequently followed by a significant decrease at −25 °C. Subsequently, there was a rapid increase in proline levels with further temperature decrease, reaching its peak at −35 °C. Dunhuang Dong Tao had the highest content of 1028.36 μg/g, while Hong Hua Shan Tao had a content of 449.73 μg/g (Figure 2C–E).

Turning to antioxidant enzyme activities, we closely examined enzymes like SOD, POD, and CAT. The activities of SOD and POD generally increased as temperatures decreased. Specifically, SOD activity across all resources remained relatively stable at temperatures above −20 °C but saw a notable increase afterward. Most of these resources reached peak SOD activity at −25 °C. Conversely, POD activity initially increased up to −20 °C, took a slight dip at this temperature, and then continued its upward trajectory. CAT activity initially increased, peaking for most resources at −10 °C. However, as temperatures dropped further, CAT activity reduced, with a minor rise observed at −25 °C, followed by a further decline as temperatures continued to plummet (Figure 3A–C).

### 3.2. LT50 of Nine Peach Resources

The relative conductivity measured under different low-temperature treatments, combined with the semi-lethal temperature (LT50) derived from the Logistic equation, provides an effective metric for understanding a plant’s sensitivity and adaptability to cold stress. Fundamentally, a lower LT50 value indicates greater cold hardiness in the given plant.

In our comprehensive analysis of branches from nine unique peach resources, the LT50 ranged from −20.55 °C to −28.13 °C. Importantly, the lowest coefficient of determination (R^2^) observed was 0.79, highlighting a strong positive correlation between relative electrical conductivity and the LT50, with this correlation being statistically significant at *p* < 0.05 (Table 4). This variation in cold hardiness among different resources emphasizes the genetic and physiological diversity within the peach species, which has significant implications for agricultural practices in diverse climate zones.

### 3.3. The Correlation Analysis of Physiological and Biochemical Indices with LT50

The LT50, determined using the electrical conductivity method, stands out as a crucial parameter for understanding and evaluating cold resistance in peach trees. We used SPSS Statistics 24.0 software to conduct an extensive correlation analysis on eight physiological and biochemical metrics: REC, SS, SP, MDA, Pro, POD, SOD, and CAT.

Our results demonstrated that the LT50 displayed a significant positive correlation with both the ELI and MDA content, with correlation coefficients of 0.894 and 0.863, respectively, which were statistically significant at *p* < 0.01. On the contrary, the SS, SP, and Pro contents showcased marked negative correlations with the LT50, with coefficients being −0.894, −0.721, and −0.863, respectively (*p* < 0.01). A significant negative correlation was also detected between the LT50 and CAT activity (*p* < 0.05) with a coefficient of −0.529. Interestingly, there was no significant correlation found with the activities of the POD and SOD enzymes (*p* > 0.05).

These findings suggest that among the indicators evaluated, ELI, SS, SP, MDA, Pro, and CAT possess strong potential as metrics for assessing cold resistance in peach trees. Additionally, the intricate relationships and correlations among these varied indicators reinforce the idea that adopting a comprehensive multi-faceted approach, rather than solely focusing on a single metric, provides a more precise assessment of the cold hardiness in peach resources (Table 5). The ELI, SS, SP, MDA, and Pro indices exhibiting highly significant correlations were chosen for subsequent resource evaluation in this study.

### 3.4. Relationship between Annual Branch Structure and Cold Resistance

The anatomical structure of peach branches comprises distinct layers: the cork layer, cortex, phloem, xylem, and pith. While this fundamental structural framework is consistent across the nine evaluated peach resources, there are noteworthy differences in the respective widths of these layers (Figure 4). Notably, the Hong Hua Shan Tao variety had the thinnest cortical layer, measuring only 84.18 μm. Yet, it possessed the widest xylem at 800.77 μm and the most substantial cork layer at 70.97 μm. Additionally, this variety recorded the highest values for XR (44.77%) and X/C (9.51). In contrast, the Qingpi Liguang Tao variety had the thickest cortical layer at 262.10 μm but showed the lowest XR and X/C values of 20.68% and 1.26, respectively (Table 6).

A comprehensive correlation analysis, relating anatomical indices to the LT50 for one-year branches (Table 7), yielded several insights. The LT50 showed significant negative correlations with the xylem thickness (X_2_), cork layer thickness (X_4_), cork layer ratio (X_8_), and X/C (X_9_), with correlation coefficients of −0.694, −0.741, −0.822, and −0.814, respectively (all statistically significant at *p* < 0.01). Additionally, the LT50 had a strong negative correlation with the xylem ratio (X_6_), with a coefficient of −0.678, and a significant positive correlation with the cortex ratio (X_7_) at 0.657 (*p* < 0.05). However, no clear correlation emerged between the cold resistance of branches and metrics like the thickness of the cortex, phloem, and the overall branch radius (*p* > 0.05). This indicates that certain anatomical characteristics might play a more critical role than others in determining the cold resistance of peach branches. The thickness of the xylem, the thickness of the cork layer, the ratio of the cork layer, and the X/C ratio all exhibit significant correlations with cold resistance. However, these data reflect overlapping information. Therefore, for the subsequent evaluation of cold resistance, we have selected the cork layer ratio and X/C ratio due to their higher significance.

### 3.5. Evaluating Cold Resistance of Peach Resources Using Mean Membership Function and Clustering Analysis

To gain insight into the cold resistance of peach resources, we analyzed 28 peach resources (varieties) subjected to a −25 °C treatment, employing six pivotal physiological and biochemical indices and two anatomical structural indices of branches linked to cold resistance. Each index’s membership function value was meticulously calculated, and the average membership function methodology was then applied to gauge the cold-resistance attributes of these resources. Upon computation, cold resistance was categorized into five distinct levels based on the membership function values, which ranged from 0.17 to 0.61 for the 28 peach resources (Table 8). Dingjiaba Liguang Tao stood out with an average membership value of 0.61, positioning it in the high-cold-resistance (HR) category; six peach germplasms fell under the cold-resistant (R) label with average membership scores between 0.50 and 0.52; eleven resources were designated as moderately cold-resistant (MR) with scores between 0.40 and 0.48; seven resources were identified with low cold resistance (LR), scoring between 0.30 and 0.38; and three resources, including Xiacui (which had the lowest mean membership function value at 0.17), were labeled as sensitive to cold (S). Regarding broader categorization (Figure 5), the 28 peach resources were segmented into two main clusters: Category I included sub-groups with resources like Rui Guang No. 39, Xia Cui, Jin Xiu, Liguangbolicui, Kanoiwa, Long Mi No. 9, Jin Hui, Longyoutao No. 1, and Wanshupingding Tao, all of which exhibited heightened sensitivity to cold temperatures, with Hong Hua Shan Tao uniquely positioned in its sub-group; Category II was further split into two subsets, with the first subset containing resources such as Dunhuang Youshui Tao, Linze Zi Tao, Sunyu No. 1, Suzhou Liguang Tao, Bai Liguang Tao, Dunhuang Dong Tao, and Wangjianguo No. 2 that demonstrated moderate cold resistance, and the second subset with resources like Dingjiabaxiao Liguang Tao, Qingpi Liguang Tao, Dunhuang Liguang Tao, and others known for marked cold resistance. Importantly, the insights from the average membership function largely aligned with the cluster analysis outcomes, emphasizing the robustness of these evaluation methods, with the only deviation being Hong Hua Shan Tao.

## 4. Discussion

### 4.1. Relationship between Physiological and Biochemical Indices and Cold Resistance of Peach

The electrical conductivity of plant tissues provides insights into the stability of the cytoplasmic membrane. The LT50, representing the semi-lethal temperature, is a widely accepted indicator of a plant’s cold resistance, acting as a testament to the strength of a plant’s resilience against cold stresses [41,42,43]. In the current study, the Electrical Leakage Index (ELI) exhibited an upward trend as temperatures decreased, forming an “S”-shaped curve. Intriguingly, the temperature corresponding to the “S” curve’s inflection point, approximately −25 °C, closely matched the LT50 value. Beyond this threshold, trees undergo significant physiological alterations. The tissues experience freezing, thereby resulting in freezing injuries [44].

Among the nine peach resources tested, the LT50 values oscillated between −28.13 °C and −20.55 °C. The LT50 values of the cultivated resources, hovering around −21 °C, echoed findings from prior research [45,46]. Notably, the LT50 values for native peach resources from Gansu were lower than those of the cultivated resources, suggesting enhanced cold-resistance capacities within these native variants.

Plants possess a remarkable ability to elevate their cold tolerance by shielding tissue cells from freezing injuries, primarily via osmotic adjustments involving specific substances [47,48]. Soluble sugars and soluble proteins, pivotal components in plant physiology, are crucial players in fortifying plant defenses against cold stresses [49,50]. Our findings revealed that the concentration of soluble sugars peaked at −25 °C, aligning with the first inflection point of the ELI. This likely indicates the activation of sugar metabolic pathways, producing protective substances that bolster cold resilience. Such accumulated sugars might function not only as osmoprotectants but also mediate cellular membrane protection, counteracting potential freeze-induced disruptions in the lipid bilayer [51,52]. Furthermore, as temperatures decreased to −30 °C, soluble protein concentrations reached their zenith. At these temperatures, it is plausible that specialized proteins, including late embryonic development-rich proteins (LEA), antifreeze proteins (AFPs), and molecular chaperones, among others, emerge, thus escalating soluble protein concentrations [53,54,55]. Under LT stress, proline can not only participate in osmotic balance and maintain membrane structure and protein stability but can also scavenge ROS [56]. In this study, the proline content fluctuated under cold stress, which was the same as the trend of proline accumulation in *Paeonia* [57] and amygdala [58] under LT stress. Interestingly, proline concentrations exhibited a sharp upswing when temperatures descended below −25 °C. This suggests that under extreme cold conditions, proline accumulation might act as peach trees’ primary protective mechanism—a conclusion that dovetails with Zhang et al.’s observations [59]. While these three osmotic regulators significantly positively correlated with the LT50 of peach branches, their periods of action varied.

Low-temperature stress can induce an increase in ROS content in plants, and with the deepening of stress, the accumulation of ROS exceeds the range that plants can tolerate, leading to adverse effects [60]. In this study, CAT activity increased rapidly at −10 °C and then decreased slowly; the trend of CAT activity change was the same as that of *Echinacea* under LT induction [61], and both peaked at LT stress. SOD decreased after rising at −10 °C, and then rose rapidly to reach a peak at −25 °C, while POD gradually increased with the decrease in temperature. The measurements of apple [12,62] and almond [58] branches at different LTs showed that MDA and POD increased first and then decreased. Under different LT stresses, the MDA content of *Zanthoxylum armatum* showed an increasing trend, and the enzyme activities of CAT, SOD, and POD showed a first increasing and then decreasing trend [63]. After LT stress, the enzyme activities of CAT, SOD, and POD in cacao germplasm with low cold resistance showed a downward trend and low activity, while those with strong cold resistance showed an upward trend and high activity [64], indicating that different species had different changes in enzyme activities under LT induction. In this study, the significant decrease in MDA content under −25 °C LT stress may be due to the increase in antioxidant enzyme activity and proline. When the temperature is lowered beyond a certain range, although the activities of SOD and POD are very high, MDA accumulation cannot be reduced due to severe damage to cells caused by extreme low temperature, and ROS scavenging enzymes cannot play a protective role. However, in this experiment, the correlation between the activities of the three enzymes and the LT50 at −25 °C was not extremely significant, which may be due to the different periods in which different enzymes play a key role. Relying on a single metric to gauge cold resistance is likely to yield an incomplete picture. However, a more holistic approach, integrating concentrations of each substance at semi-lethal temperatures, offers a more nuanced and accurate evaluation of cold resilience in peach resources.

### 4.2. Relationship between Branch Anatomical Structure and Cold Resistance of Peach

Plants, when exposed to environmental challenges, undergo numerous changes across morphological, physiological, biochemical, and anatomical facets [65,66]. These adaptations strengthen their resilience, shielding them from potential disturbances [23]. Consequently, anatomical traits have emerged as vital markers in cold-resistance research, acting as critical reference points [24,67].

The xylem, predominantly made up of cells with low activity, is crucial for water transportation. The vessel density and diameter within the xylem directly impact its efficiency, influencing a plant’s cold resistance. Our study found a direct correlation between the phellem and xylem thicknesses and a branch’s cold resistance. Although there were variances in the xylem and cortex tissue thicknesses within samples of the same variety, their ratios in branches were consistent. In this study, the membership function degree of the CLR and X/C of the rootstock Hong Hua Shan Tao was the highest. This relation between the xylem and cortex in terms of cold resistance highlights the intrinsic structural traits of peach resources related to cold resistance, echoing Wang’s conclusions [68]. Cold-resistant resources notably have a more defined phellem layer and a higher degree of phellem transformation, consistent with findings from walnut branch research [69]. The phloem thickness of Kanoiwa was significantly higher than that of other varieties in this study; however, its cold resistance was found to be low (LR) in the comprehensive evaluation, which contradicts previous findings on the cold resistance of raspberry [67] and apple [70]. Anatomical features can provide considerable insight into a plant’s cold resistance, offering valuable comparative measures for different resources. However, a comprehensive assessment of cold hardiness should include physiological and biochemical markers, as well as observations of field damage and post-damage recovery. Our combined analysis of the mean membership function values and clustering unveiled that Gansu’s indigenous resources have superior cold resistance compared to cultivated ones. Interestingly, native resources grouped together, emphasizing the enhanced cold resilience of Gansu resources. Conversely, cultivated types seem to have developed reduced cold resistance over time compared to indigenous ones, possibly a result of the natural selection process. This aligns with Wang’s analysis of 58 Chinese peach resources [45].

Within the Gansu Hexi Corridor, although winter temperatures fluctuate across areas, there is a remarkable consistency among introduced regions. This homogeneity might be due to extensive human interactions or the deep genetic connections established over extensive periods. As a result, distinguishing cold resistance in peach resources from different Gansu areas becomes challenging.

### 4.3. Evaluation of Cold Resistance of Gansu Local Peach Resources using Membership Function

The complexity of plant cold resistance arises from the synergistic interplay of genetics, environment, and physiology. Rather than being a singular trait, cold resistance is quantitative, orchestrated by a complex array of genes throughout a plant’s genome. Understanding this intricacy is vital when evaluating the cold resistance of peach resources. Relying solely on one metric oversimplifies this multifaceted trait, potentially missing the true depth of a plant’s resilience to cold. A more encompassing methodology, which considers multiple indices and then derives an average membership function using a specific formula, provides a more thorough understanding of cold resistance, enhancing both the reliability and validity of evaluations [71,72].

In our comprehensive study that integrated physiological, biochemical, and anatomical aspects, we assessed 28 peach resources. A key metric, the LT50, was compared with results from the average membership function. Generally, the findings were in sync, but discrepancies surfaced in certain resources. For instance, Dingjiaba displayed exceptional cold resilience using both evaluation methods. On the other hand, Honghuashantao presented a dichotomy: while its physiological and biochemical indices hinted at diminished cold tolerance, its anatomical markers suggested the opposite. Such variations might be attributed to intrinsic differences between Honghuashantao and other peach resources under study. Although the LT50 is fundamental for cold-resistance evaluation, its effectiveness is heightened when combined with other measures. An integrated evaluation method reduces biases associated with singular metrics, offering a more nuanced view of a plant’s genuine cold-resistance capabilities [25].

A salient observation from our research underscores the superior cold resistance of Gansu’s indigenous peach resources in comparison to cultivated resources, showcasing their resilience forged over countless years of natural selection. This aligns with Wang et al.’s comprehensive study of 58 unique Chinese peach resources [45].

Exploring the Gansu Hexi Corridor reveals a diversity of winter temperatures. However, there is a commonality: the cold resistance exhibited in different introduced regions is strikingly consistent. This consistency could result from persistent human interactions or from the entrenched genetic links nurtured over extended periods of crossbreeding and exchanges. This uniformity makes distinguishing cold-resistance levels amongst peach resources from various Gansu regions challenging.

## 5. Conclusions

This study provided a comprehensive assessment of the physiological, biochemical, and anatomical responses of peach resources under low-temperature stress. It is evident that different peach resources display marked differences in cold hardiness, primarily attributed to their distinct physiological and biochemical mechanisms. The LT50 values, MDA content, and the enzymatic activities of SOD, POD, and CAT offer profound insights into the cold-stress responses of these peach resources. Notably, anatomical structures, especially the thicknesses of xylem and cork layers, play a significant role in the cold resistance of peach trees. By evaluating the cold resistance of 28 peach resources, we further pinpointed which resources stand robust under chilly conditions. The average membership function method and cluster analysis presented a consistent and reliable framework for categorizing and evaluating the cold-resistance attributes of peach resources. Overall, these findings offer invaluable insights into the cold-hardiness of peach trees and provide guidance for agricultural practices, especially when selecting peach resources adapted to specific cold environments. However, the cold resistance of peach trees is influenced by various factors and not solely determined by the lowest winter temperature. Moreover, Gansu Hexi Corridor peaches exhibit superior resilience compared to cultivated varieties during springtime. Consequently, our future research will focus on investigating the hibernation process of these cold-resistant resources.

## Figures and Tables

**Figure 1 plants-12-04183-f001:**
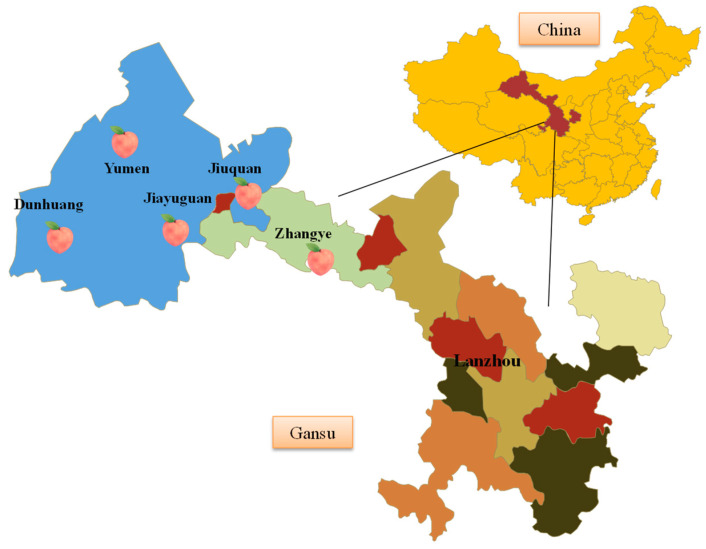
Sampling diagram of local peach resources from the Hexi Corridor in Gansu province.

**Figure 2 plants-12-04183-f002:**
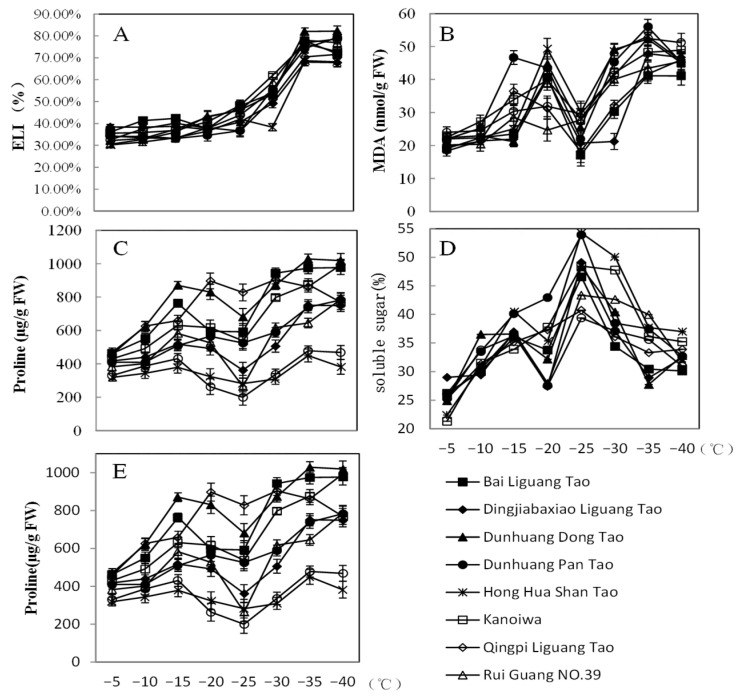
Changes in physiological and biochemical indices of 9 peach resources under varying LTs. (**A**): ELI; (**B**): MDA; (**C**): Proline; (**D**): Soluble sugar; (**E**): Soluble protein. Error bars denote standard errors.

**Figure 3 plants-12-04183-f003:**
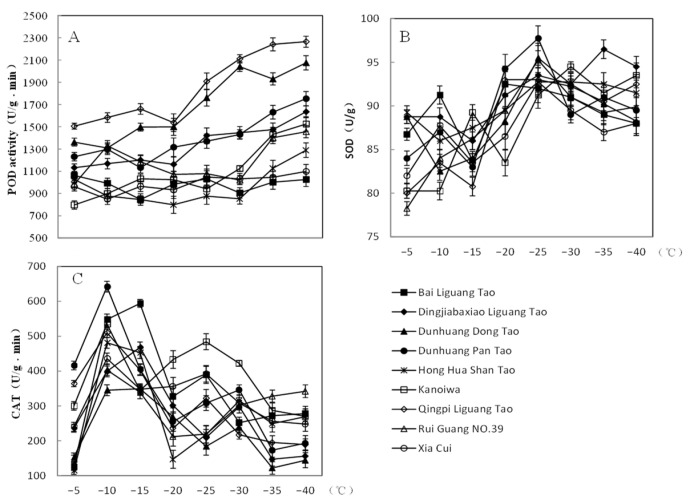
Changes in enzyme activity of 9 peach resources under varying LTs. (**A**): POD activity; (**B**): SOD activity; (**C**): CAT activity. Error bars denote standard errors.

**Figure 4 plants-12-04183-f004:**
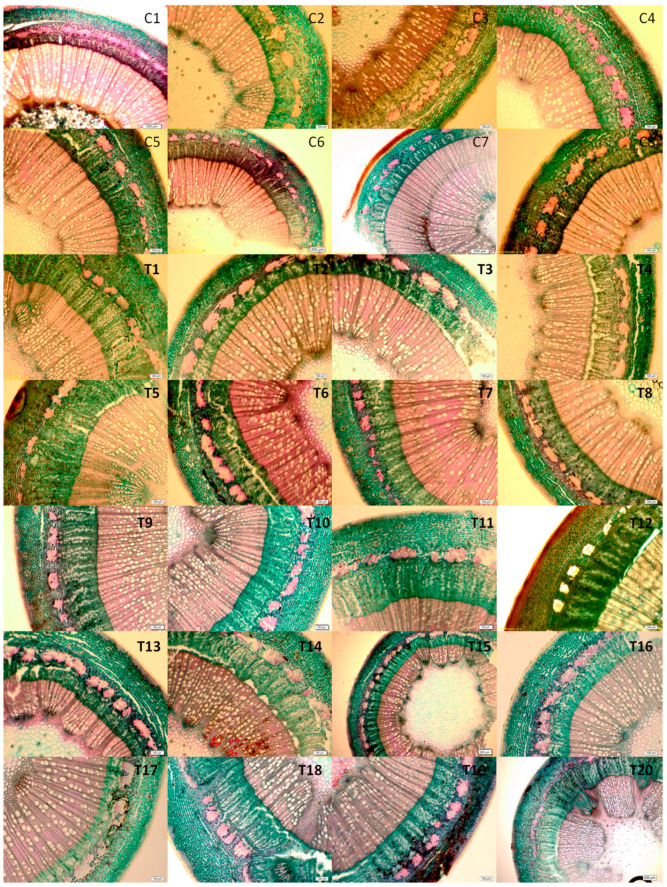
Anatomical structures of annual branches of 28 peach germplasms in dormant period.

**Figure 5 plants-12-04183-f005:**
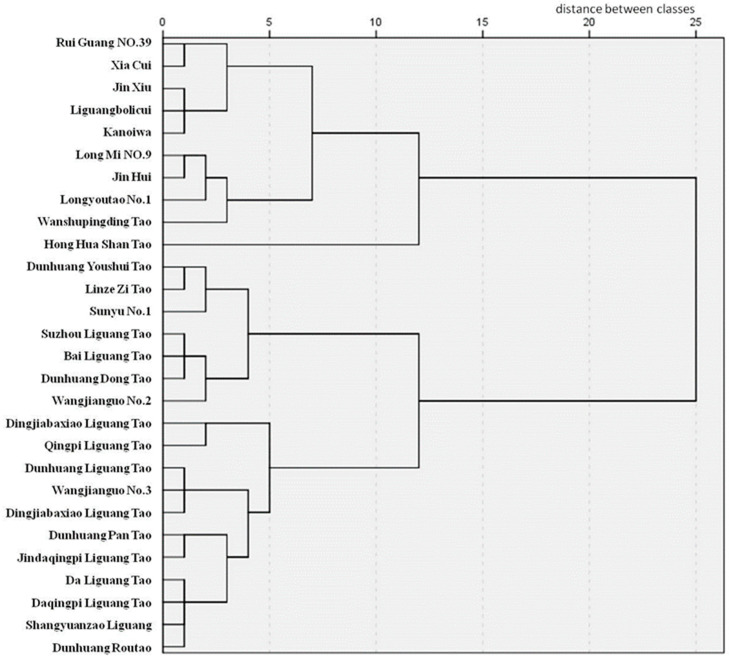
Dendrogram of hierarchical cluster analysis of pedigree of 28 peach resources.

**Table 1 plants-12-04183-t001:** The characteristics of peach resources for screening evaluation indices of cold resistance.

No.	Variety	Type	Source
1	Bai Liguang Tao	Nectarine	Jiuquan, Gansu
2	Dingjiabaxiao Liguang Tao	Nectarine	Jiuquan, Gansu
3	Dunhuang Dong Tao	Peach	Dunhuang, Gansu
4	Dunhuang Pan Tao	Flat Peach	Dunhuang, Gansu
5	Hong Hua Shan Tao	Root Stock	Gansu
6	Kanoiwa	Peach	Japan
7	Qingpi Liguang Tao	Nectarine	Yumen, Gansu
8	Rui Guang No. 39	Nectarine	Beijing
9	Xia Cui	Peach	Jiangsu

**Table 2 plants-12-04183-t002:** Characteristics of peach cultivars for evaluation.

No.	Resource	Type	Source
C1	Hong Hua Shan Tao	Root Stock	Gansu
C2	Jin Hui	Nectarine	Henan
C3	Jin Xiu	Peach	Shanghai
C4	Kanoiwa	Peach	Japan
C5	Longmi No. 9	Peach	Gansu
C6	Longyoutao No. 1	Nectarine	Gansu
C7	Rui Guang No. 39	Nectarine	Beijing
C8	Xia Cui	Peach	Jiangsu

**Table 3 plants-12-04183-t003:** Characteristics of Gansu local peach germplasm resources for evaluation.

No.	Germplasm	Peach Type	Source
T1	Bai Liguang Tao	Nectarine	Dunhuang
T2	Da Liguang Tao	Nectarine	Jiayuguan
T3	Daqingpi Liguang Tao	Nectarine	Jiuquan
T4	Dingjiaba Liguang Tao	Nectarine	Jiuquan
T5	Dingjiabaxiao Liguang Tao	Nectarine	Jiuquan
T6	Dunhuang Dong Tao	Peach	Dunhuang
T7	Dunhuang Pan Tao	Flat Peach	Dunhuang
T8	Dunhuang Liguang Tao	Nectarine	Dunhuang
T9	Dunhuang Routao	Nectarine	Dunhuang
T10	Dunhuang Youshui Tao	Nectarine	Yumen
T11	Jindaqingpi Liguang Tao	Nectarine	Jiuquan
T12	Liguangbolicui	Nectarine	Dunhuang
T13	Linze Zi Tao	Peach	Zhangye
T14	Qingpi Liguang Tao	Nectarine	Jiuquan
T15	Shangyuanzao Liguang	Nectarine	Dunhuang
T16	Sunyu No. 1	Nectarine	Jiuquan
T17	Suzhou Liguang Tao	Nectarine	Jiuquan
T18	Wangjianguo No. 2	Nectarine	Jiuquan
T19	Wangjianguo No. 3	Nectarine	Dunhuang
T20	Wanshupingding Tao	Nectarine	Jiuquan

**Table 4 plants-12-04183-t004:** Logistics equation of the relative electrical conductivity and the LT50 of 9 peach resources.

Variety	Logistics Equation	LT50/°C	R^2^	Sequence of Cold Resistance
Hong Hua Shan Tao	Y = 100/(1 + 3.93e^−0.0485x^)	−28.13	0.85	1
Qingpi Liguang Tao	Y = 100/(1 + 3.79e^−0.0515x^)	−25.87	0.87	2
Dingjiabaxiao Liguang Tao	Y = 100/(1 + 2.79e^−0.0401x^)	−25.63	0.82	3
Dunhuang Dong Tao	Y = 100/(1 + 4.09e^−0.0634x^)	−24.11	0.86	4
Dunhuang Pan Tao	Y = 100/(1 + 4.58e^−0.0632x^)	−24.08	0.86	5
Xia Cui	Y = 100/(1 + 4.02e^−0.0621x^)	−22.39	0.93	6
Rui Guang No. 39	Y = 100/(1 + 4.22e^−0.0651x^)	−22.11	0.79	7
Bai Liguang Tao	Y = 100/(1 + 3.34e^−0.0547x^)	−21.86	0.81	8
Kanoiwa	Y = 100/(1 + 3.35e^−0.0561x^)	−20.55	0.85	9

**Table 5 plants-12-04183-t005:** Correlation analysis of cold-resistance indices of different peach resources.

Indices	X_1_	X_2_	X_3_	X_4_	X_5_	X_6_	X_7_	X_8_	LT50
X_1_	1								
X_2_	0.870 **	1							
X_3_	−0.882 **	0.855 **	1						
X_4_	−0.788 **	−0.698 *	−0.754 **	1					
X_5_	−0.736 **	0.873 **	0.877 **	−0.974 **	1				
X_6_	−0.335	−0.420	−0.587	0.240	−0.581	1			
X_7_	−0.235	0.302	0.543	−0.231	0.520	0.754	1		
X_8_	−0.510 *	0.324	0.453	−0.440	0.232	0.432	0.550	1	
LT50	0.894 **	−0.874 **	−0.721 **	0.742 **	−0.863 **	−0.369	−0.323	−0.529 *	1

Note: * represents the significance level of *p* < 0.05; ** represents the significance level of *p* < 0.01. The variables X_1_–X_8_ represent 8 physiological and biochemical indicators (X_1_: ELI, X_2_: SS, X_3_: SP, X_4_: MDA, X_5_: Pro, X_6_: POD, X_7_: SOD, X_8_: CAT).

**Table 6 plants-12-04183-t006:** Annual branch characteristics of 9 peach cultivars.

Resource No.	Cortical (μm)	Xylem (μm)	Phloem (μm)	Cork Layer (μm)	Radius (μm)	XR (%)	CR (%)	CLR (%)	X/C
Bai Liguang Tao	154.39 ± 6.42 b	494.24 ± 7.03 b	276.01 ± 4.73 bc	10.53 ± 0.95 e	1470.92 ± 102.65 c	33.60 ± 4.65 b	10.50 ± 1.02 bc	0.72 ± 0.11 e	3.20 ± 0.27 c
Dingjiabaxiao Liguang Tao	150.45 ± 13.08 b	392.27 ± 9.74 c	269.34 ± 21.91 bc	26.40 ± 4.75 b	1335.03 ± 112.44 c	29.38 ± 3.22 bc	11.27 ± 0.88 b	1.98 ± 0.26 b	2.61 ± 0.34 cd
Dunhuang Dong Tao	118.96 ± 16.12 c	554.60 ± 24.1 b	217.25 ± 28.00 bc	14.95 ± 4.80 d	1383.34 ± 98.78 c	40.09 ± 4.18 a	8.60 ± 0.76 c	1.08 ± 0.08 de	4.66 ± 0.52 b
Dunhuang Pan Tao	236.90 ± 27.48 a	352.83 ± 14.33 bc	367.16 ± 25.15 b	21.25 ± 7.32 c	1640.31 ± 133.45 b	21.51 ± 3.66 d	14.44 ± 1.23 b	1.30 ± 0.12 cd	1.49 ± 0.11 e
Hong Hua Shan Tao	84.18 ± 7.29 e	800.77 ± 31.88 a	223.49 ± 16.56	70.97 ± 6.81 a	1788.46 ± 133.29 a	44.77 ± 4.72 a	4.71 ± 0.52 d	3.97 ± 0.43 a	9.51 ± 1.26 a
Kanoiwa	164.14 ± 17.65 b	489.03 ± 14.48 b	519.45 ± 73.61 a	26.76 ± 9.11 b	1840.74 ± 141.62 a	26.57 ± 2.86 c	8.92 ± 0.87 c	1.45 ± 0.25 c	2.98 ± 0.27 c
Qingpi Liguang Tao	262.10 ± 6.50 a	331.09 ± 3.29 cd	362.71 ± 12.23 b	25.79 ± 1.86 b	1600.73 ± 125.65 b	20.68 ± 2.34 d	16.37 ± 1.42 a	1.61 ± 0.21 bc	1.26 ± 0.08 e
Rui Guang No. 39	133.50 ± 12.84 bc	284.48 ± 7.07 d	222.74 ± 3.91 bc	16.01 ± 2.93 d	1145.83 ± 89.56 d	24.83 ± 3.05 cd	11.65 ± 1.65 b	1.40 ± 0.14 c	2.13 ± 0.43 d
Xia Cui	240.89 ± 9.09 a	509.47 ± 21.54 b	212.25 ± 11.21 bc	28.81 ± 4.37 b	1786.84 ± 110.43 a	28.51 ± 1.97 bc	13.48 ± 1.77	1.61 ± 0.15 bc	2.11 ± 0.29 d
mean	171.72	467.64	296.71	26.83	1554.69	29.99	11.10	1.68	3.33

Note: Distinct letters signify statistically significant differences between means at the *p* < 0.05 level.

**Table 7 plants-12-04183-t007:** Correlative matrix of branch structure parameters and LT50.

Indices	X_1_	X_2_	X_3_	X_4_	X_5_	X_6_	X_7_	X_8_	X_9_	LT50
X_1_	1									
X_2_	−0.400 *	1								
X_3_	0.470 *	−0.140	1							
X_4_	−0.140	0.483 *	0.024	1						
X_5_	0.404 *	0.493 **	0.425 *	0.552 **	1					
X_6_	−0.639 **	0.887 **	−0.347	0.209	0.048	1				
X_7_	0.842 **	−0.706 **	0.315	−0.414 *	−0.121	−0.726 **	1			
X_8_	−0.166	0.414 *	−0.102	0.909 **	0.411 *	0.209	−0.386 *	1		
X_9_	−0.712 **	0.842 **	−0.300	0.631 **	0.200	0.824 **	−0.845 **	0.562 **	1	
LT50	0.340	−0.694 **	0.269	−0.741 **	−0.134	−0.678 *	0.657 *	−0.822 **	−0.814 **	1

Note: * represents the significance level of *p* < 0.05; ** represents the significance level of *p* < 0.01. The variables X_1_–X_9_ represent 9 indicators of branches anatomical structure (X_1_: cortical thickness, X_2_: xylem thickness, X_3_: phloem thickness, X_4_: cork layer thickness, X_5_: branch radius, X_6_: xylem ratio, X_7_: cortex ratio, X_8_: cork layer ratio, X_9_: X/C).

**Table 8 plants-12-04183-t008:** Identification results of cold resistances of 28 peach germplasm resources.

Variety	ELI	MDA	Pro	SP	SS	CLR	X/C	Mean Membership Function Value	Cold Resistance Level
Dingjiaba Liguang Tao	1.00	1.00	0.69	0.55	0.44	0.33	0.25	0.61	HR
Qingpi Liguang Tao	0.92	0.92	0.93	0.52	0.06	0.00	0.28	0.52	R
Hong Hua Shan Tao	0.49	0.00	0.13	0.28	0.70	1.00	1.00	0.51	R
Dingjiabaxiao Liguang Tao	0.60	0.86	0.25	0.83	0.45	0.16	0.39	0.51	R
Dunhuang Pan Tao	0.91	0.63	0.49	0.63	0.68	0.03	0.18	0.51	R
Dunhuang Youshui Tao	0.26	0.46	0.80	1.00	0.46	0.17	0.39	0.51	R
Shangyuanzao Liguang	0.62	0.89	0.69	0.60	0.37	0.16	0.15	0.50	R
Jindaqingpi Liguang Tao	0.63	0.63	0.55	0.64	0.70	0.09	0.14	0.48	MR
Sunyu No. 1	0.35	0.49	0.57	0.73	0.13	0.34	0.56	0.45	MR
Linze Zitao	0.35	0.54	1.00	0.83	0.25	0.01	0.19	0.45	MR
Dunhuang Liguang Tao	0.57	0.91	0.22	0.71	0.22	0.39	0.13	0.45	MR
Wangjianguo No. 3	0.44	0.84	0.47	0.76	0.22	0.29	0.06	0.44	MR
Da Liguang Tao	0.36	0.84	0.49	0.35	0.65	0.27	0.02	0.43	MR
Suzhou Liguang Tao	0.06	0.64	0.73	0.94	0.39	0.13	0.10	0.43	MR
Dunhuang Routao	0.57	0.72	0.51	0.32	0.44	0.06	0.33	0.42	MR
Daqingpi Liguang Tao	0.25	0.71	0.52	0.54	0.41	0.18	0.23	0.41	MR
Bai Liguang Tao	0.02	0.76	0.68	0.65	0.49	0.23	0.00	0.41	MR
Dunhuang Dong Tao Longmi No. 9	0.16 0.88	0.36 0.33	0.72 0.07	0.63 0.22	0.42 0.80	0.41 0.18	0.11 0.14	0.40 0.38	MR LR
Wanshupingding Tao	0.48	0.06	0.56	0.51	0.59	0.08	0.24	0.36	LR
Jin Xiu	0.59	0.52	0.21	0.36	0.37	0.15	0.17	0.34	LR
Longyoutao No. 1	0.30	0.39	0.19	0.07	1.00	0.23	0.08	0.32	LR
Liguangbolicui	0.47	0.32	0.34	0.41	0.22	0.17	0.27	0.31	LR
Wangjianguo No. 2	0.55	0.73	0.09	0.49	0.07	0.15	0.11	0.31	LR
Kanoiwa	0.00	0.54	0.51	0.21	0.42	0.21	0.23	0.30	LR
Jin Hui	0.64	0.32	0.00	0.09	0.41	0.06	0.16	0.24	S
Rui Guang No. 39	0.18	0.44	0.10	0.06	0.19	0.11	0.21	0.18	S
Xia Cui	0.35	0.42	0.01	0.00	0.00	0.10	0.28	0.17	S

## Data Availability

Data are contained within the article.

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
