# Peer review of "Physiochemical Responses and Ecological Adaptations of Peach to Low-Temperature Stress: Assessing the Cold Resistance of Local Peach Varieties from Gansu, China"

_plants, 2023, doi:10.3390/plants12244183_

Round 1

Reviewer 1 Report

Comments and Suggestions for Authors

The study investigated cold resistance of nine peach varieties using physiological, biochemical, and anatomical. The study is well designed but need some revisions which are mentioned in my comments to authors.

The abstract is poorly written mainly lacking short details of methods and background of the study.

“played diverse patterns in response to temperature changes” clarify and specify.

“identified six pivotal indicators, inclusive of” also mention the significant results.

Line number is missing.

Small letters “ Local Peach Resources” for L and R if this is not a specific term or name.

As the study deals with “Physiochemical Responses and Ecological Adaptations” so these aspects must be clearly discussed in the introduction.

Introduction is very short and information or literature review is not sufficient. The topic has been progressively investigated so include different aspects specifically the following points.

Paragraph 1 mention about biochemical pathways, specific hormones related to cold stress.

“cold resistance and attributes like xylem thickness” should be clarify.

Paragraph 2 add economic importance of the peach, specifically in terms of China. The following study could be use as reference discussing the economic importance and significant of Peach.  https://doi.org/10.1016/j.bcab.2020.101729

Also add cold challenges to Peach production and harms occurring to Peach annually.

Would be better to add a map of the sampling sites in section 2.1

Section 2.2 provide references

Also provide references in section 2.3 the following work on microscopic studies would be better to cite. https://doi.org/10.3390/agronomy13010269, https://doi.org/10.3390/agronomy12102500.

“Plants, when exposed to environmental challenges, undergo numerous changes across morphological, physiological, biochemical, and anatomical facets” the sentence should be cited with some recent and relevant studies. The following studies may be helpful. https://doi.org/10.1016/j.agrformet.2023.109734, https://doi.org/10.1016/j.scitotenv.2022.160930

Section 4.1 discuss the mechanism affecting the biochemical process.

Which environmental effects have been examined in this study to justify ecological adaptation.

Add future perspective in the conclusion

Comments on the Quality of English Language

English changes are mentioned in comments to authors

Author Response

Summary

Thank you very much for taking the time to review this manuscript. Your reviews have been very instructive to this article. Based on your question, please find the detailed responses below and the corresponding revisions changes in the re-submitted files.

Point-by-point response to Comments and Suggestions for Authors

Comments 1: [The abstract is poorly written mainly lacking short details of methods and background of the study.]

Response 1: Thank you for pointing this out. We agree with this comment. Therefore, the abstract has been rewritten “[The abstract has been updated in red in the article]”

Comments 2: [played diverse patterns in response to temperature changes” clarify and specify]  

Response 2: We clarify and specify description of the temperature change. (2.1. Plant Materials, Paragraph 1, line 8)

Comments 3: [identified six pivotal indicators, inclusive of” also mention the significant results.]  

Response 4: The reasons for the selection have been added to the text.[section 3.3 Paragraph 3, line 6][ section 3.4 Paragraph 2, line 11]

Comments 4: [Line number is missing].

Response 4: It should be that the editorial department has deleted the line number, and we have explained the modification suggestions in detail according to the part.

Comments 5: [Small letters “ Local Peach Resources” for L and R if this is not a specific term or name. ]

Response 5: We have changed the initial letter of "Local Peach Resources" in the article to small letters.

Comments 6: [As the study deals with “Physiochemical Responses and Ecological Adaptations” so these aspects must be clearly discussed in the introduction.].

Response 6: The discussion of physiochemical responses and ecological adaptations is added in the introduction

Comments 7:Introduction is very short and information or literature review is not sufficient. The topic has been progressively investigated so include different aspects specifically the following points.

1.Paragraph 1 mention about biochemical pathways, specific hormones related to cold stress.

2.“cold resistance and attributes like xylem thickness” should be clarify.

3.Paragraph 2 add economic importance of the peach, specifically in terms of China. The following study could be use as reference discussing the economic importance and significant of Peach.  https://doi.org/10.1016/j.bcab.2020.101729

4.Also add cold challenges to Peach production and harms occurring to Peach annually.

Response7: Related content is added in the introduction.

Comments 8:Would be better to add a map of the sampling sites in section 2.1

Response8: Added a sampling map(figure1).

Comments 9:

1.Section 2.2 provide references

2.Also provide references in section 2.3 the following work on microscopic studies would be better to cite. https://doi.org/10.3390/agronomy13010269, https://doi.org/10.3390/agronomy12102500.

3.“Plants, when exposed to environmental challenges, undergo numerous changes across morphological, physiological, biochemical, and anatomical facets” the sentence should be cited with some recent and relevant studies. The following studies may be helpful. https://doi.org/10.1016/j.agrformet.2023.109734,  https://doi.org/10.1016/j.scitotenv.2022.160930

Response9: Thank you for your reminding. Related references is added.

Comments 10:

1.Section 4.1 discuss the mechanism affecting the biochemical process.

2.Which environmental effects have been examined in this study to justify ecological adaptation.

3.Add future perspective in the conclusion

Response10:

  1. The discussion of mechanism affecting the biochemical process was added
  2. In this paper, the organizational structure of branches is used as the index of peach's long-term adaptation to the environment, and the specific environmental index is not involved in the research. In future studies, we will study the relationship between environmental factors and organizational structure indicators.
  3. Future perspective added in the conclusion(section 5. line 6)

Reviewer 2 Report

Comments and Suggestions for Authors

The manuscript is presenting huge data with sound scientific methodologies. It is also written well and providing a well framed conclusion. However, I wonder why the molecular data is missing. I wish to see the molecular analysis to provide the full picture of this tolerance. 

Also, many recent references are missing in the discussion section.

Comments on the Quality of English Language

Minor issues. 

Author Response

Summary

Thank you very much for taking the time to review this manuscript. Your reviews have been very instructive to this article. Based on your question, please find the detailed responses below and the corresponding revisions changes in the re-submitted files.

Comments

The manuscript is presenting huge data with sound scientific methodologies. It is also written well and providing a well framed conclusion. However, I wonder why the molecular data is missing. I wish to see the molecular analysis to provide the full picture of this tolerance.

Also, many recent references are missing in the discussion section.]

Response:

We have conducted molecular-level testing on the selected cold-resistant resources and published a referenced paper [Reference 53]. Additionally, we have recently incorporated the most up-to-date literature.

Reviewer 3 Report

Comments and Suggestions for Authors

This study investigated the cold tolerance of nine peach varieties using physiological, biochemical and anatomical criteria. From the physiological and biochemical point of view, the trends of electrolyte leakage index (ELI) and malondialdehyde (MDA) content of different peach varieties were verified, and it was found that different peroxidative enzymes responded to the temperature changes in different patterns. The article utilized conductivity to determine the half-lethal temperature (LT50) of different varieties as a means of further evaluating the plant's cold resistance, and further used anatomical perspectives to demonstrate a significant correlation between cold resistance and attributes such as xylem thickness. The experimental data were well-designed and analyzed comprehensively, and the results provide a certain reference value for selecting the most suitable local peach resources for cold-resistant breeding. However, there are still some problems that need to be solved, and they are as follows:

1. Why ELI, MDA, POD and other physiological indicators were selected, but the authors did not explain the reason for selecting physiological indicators in the results section, the language needs to be added again to increase the coherence of the context.

2. Author please check the content of all the charts carefully, there are problems with incorrect writing in some of the data, such as "-.639**" in Table 7, please check and correct it.

3. Fig 1 picture is not clear enough, it is recommended to replace it with a high-resolution figure or change it to a line graph.

4. It is suggested that words with abbreviations in the text and the figure should be labeled with abbreviations in parentheses at the back when they appear for the first time, or a separate note should be added to the figure, to prevent misinterpretation due to the incoherence of the context.

5. Regarding the proposal of LT50, the article only suggests that it is calculated using a logistic equation in the results section, so please add the relevant calculation content in the Materials and Methods section. In addition, why ELI was selected as the variable for calculation in the equation, the reason for its selection and why other physiological indicators were not selected were not clearly elaborated, please add the proof.

6What statistical method was used for cluster analysis in Fig 3? What is the database source for the comparison? Please add in Materials and Methods and Fig 3 figure notes.

7Most of the literature cited by the authors is before 18 years and lacks research progress in the last 5 years. It is recommended that the authors read and add 5-8 references.

Comments on the Quality of English Language

The introduction and discussion are well-written, and some contextual coherence can be added to the results section.

Author Response

Summary

Thank you very much for taking the time to review this manuscript. Your reviews have been very instructive to this article. Based on your question, please find the detailed responses below and the corresponding revisions changes in the re-submitted files.

Comments 1. Why ELI, MDA, POD and other physiological indicators were selected, but the authors did not explain the reason for selecting physiological indicators in the results section, the language needs to be added again to increase the coherence of the context.

Response 1: The justifications for the selection have been incorporated into the text. [section 3.3 Paragraph 3, line 6] [section 3.4 Paragraph 2, line 11]

Comments 2. Author please check the content of all the charts carefully, there are problems with incorrect writing in some of the data, such as "-.639**" in Table 7, please check and correct it.

Response 2: The comments are greatly appreciated. All the data has been thoroughly reviewed and rectified.

Comments 3. Fig 1 picture is not clear enough, it is recommended to replace it with a high-resolution figure or change it to a line graph.

Response 3: The graph transformed into a line graph. [figure2]

Comments 4. It is suggested that words with abbreviations in the text and the figure should be labeled with abbreviations in parentheses at the back when they appear for the first time, or a separate note should be added to the figure, to prevent misinterpretation due to the incoherence of the context.

Response 4: The initial appearances of abbreviations are elaborated upon in detail.

Comments 5. Regarding the proposal of LT50, the article only suggests that it is calculated using a logistic equation in the results section, so please add the relevant calculation content in the Materials and Methods section. In addition, why ELI was selected as the variable for calculation in the equation, the reason for its selection and why other physiological indicators were not selected were not clearly elaborated, please add the proof.

Response 5: The use of ELI as a variable is deemed more reliable, given our extensive referencing of previous methodologies. Additionally, we have included relevant citations within the paper. It should be noted that the logistic equation calculation method, being widely employed, does not warrant repetition in this study.

Comments 6What statistical method was used for cluster analysis in Fig 3? What is the database source for the comparison? Please add in Materials and Methods and Fig 3 figure notes.

Response 6: The cluster analysis in Fig 3 was conducted using a statistical method (Section 2.4, Paragraph 1, Line 5).

Comments 7Most of the literature cited by the authors is before 18 years and lacks research progress in the last 5 years. It is recommended that the authors read and add 5-8 references.

Response 7: The current and pertinent literature is supplemented.

Reviewer 4 Report

Comments and Suggestions for Authors

Review Report:

Manuscript ID: plants-2722979

Physiochemical Responses and Ecological Adaptations of Peach to Low-Temperature Stress: Assessing the Cold Resistance of Local Peach Varieties from Gansu, China

Title: Title is OK

Abstract: The rootstock utilized with different peach cultivars is the most essential criterion for researching abiotic stresses in temperate fruits crops. There is no information on the rootstock/scion combo. The majority of the time, rootstocks control resistance/tolerance. 

What is the meaning of bore in ‘inclusive of ELI and MDA, that bore significant’?  Fix it.

Introduction:

1: Not flowing with language. Change the words. ‘osmotic regulating substances’.

2: Do not use fancy word like diving. ‘temperatures diving’. Fix it.

3: It is difficult to determine what causes these trees' cold resistance. Is it the result of rootstock or scion? The authors did not discuss any review of literature on factors responsible for peach cold tolerance. There was no evaluation of the research on rootstocks, how chilling harms rootstock/scion, and how chilling affects oxidative stress and antioxidant defense. The subject's introduction is brief, and necessary references for the parameters under consideration are required.

Ref:

1: Rootstock-Mediated Transcriptional Changes Associated with Cold Tolerance in Prunus mume Leaves.

2: Low-temperature susceptibility of 'redhaven' peach floral buds on various rootstocks in the 1994 NC-140 trial

3: Transcriptome profiling of Prunus persica branches reveals candidate genes potentially involved in freezing tolerance.

4: A comparison of seasonal ultrastructural changes in stem tissues  of peach (Prunus persica) that exhibit contrasting mechanisms of cold hardiness.

Method:

 1: No need to use fancy word like renowned. ‘renowned Peach Genetic Repository’

2: ‘Samples, comprising 28 distinct germplasms’. Germplasm in what form was requested?

3: ‘On January 5th’. What year?

4: Only rootstock was the part of this study. What is the significance of this work when cold hardiness is governed by a rootstock in a rootstocks/scion combination? In fact, rootstock has the ability to alter the physiology and molecular biology of scions. These points must be clarified by the authors. 

Physiological and Molecular Aspects of Plant Rootstock-Scion Interactions. https://www.frontiersin.org/articles/10.3389/fpls.2022.852518/full

5: How many replicates are there per scion (the central section of a branch)? Only three replicates per treatments? How many times was the experiment repeated? One or two times with three duplicates of the middle section of the branch per treatment? N =? Provide this information in a clear manner.

6: Why was no review provided in the introductory section for the following parameters? A review will assist the reader in better understanding the consequences of these parameters?

CR (Cortex ratio of branches) = (thickness of cortex/branch radius) × 100%; CLR (Cork layer ratio) = (cork layer thickness/branch radius) × 100%; XR (Xylem ratio) = (xylem thickness/branch radius) × 100%; X/C = xylem thickness/cortex thickness.

Results:

1: At 300% resolution, a graph looks like this. Data visualization is tricky. I delegated the decision on the display clarity of these graphs to the editor. Nothing is visible at 100% resolution.

2: ‘For the nine varieties'. One cannot argue that there are nine varieties, one of which is rootstock and the others are scions.  

3: Generalized data were reported with no consideration for the scions and rootstocks employed. It is nearly impossible to visualize any conclusion using graphs. Authors must report results for each scion and rootstock, as well as how data changes as temperature decreases. This image would benefit breeders rather than the general results.

4: Let me ask a question: at what temperature did the authors find the maximum and lowest proline concentrations, and in which scion and rootstock? Such material in the results section will assist readers better understand the findings of the investigation. 

5: “Considering specific varieties, "Hong Hua Shan Tao" exhibited an LT50 value below -28℃, identifying it as exceptionally cold-tolerant. Conversely, the "Kanoiwa" variety recorded an LT50 of -20.55℃, marking it as the least cold-hardy among the examined peach cultivars”

This is a technical issue. Why should a rootstock (Hong Hua Shan Tao) be compared to a scion (Kanoiwa)? Alternatively, with other scions. Rootstocks must only be compared to other rootstocks. Why so? Because no one investigates rootstock branches. It is the root that is important. However, in scion, branches are important because they bear fruit.

6: ‘boasting correlation coefficients’. Do use fancy word like boasting.

Discussion:

1: ‘forming an "S"-shaped curve’. I did not see any such graph in the manuscript. Only bar diagrams.

2: For proline, cat, pod, and sod, there is only generalized discussion with no pertinent references. There is no information or discussion about the selected rootstocks and scions. Surprisingly, the authors did not address antioxidant defense factors in the discussion section.

3: Zhan et al.'s observations [33]. Or Zhang? On sugarcane.

Transcriptome Analysis of Genes Involved in Cold Hardiness of Peach Tree (Prunus persica) Shoots during Cold Acclimation and Deacclimation

4: In the discussion section for anatomical research, not a single rootstock or scion was mentioned. Simply a textbook-style explanation.

I recommend resubmission after major revision, with an emphasis on clarifying technical issues.

Comments on the Quality of English Language

Minor changes. 

Author Response

1. Summary

Thank you very much for taking the time to review this manuscript. Your reviews have been very instructive to this article. Based on your question, please find the detailed responses below and the corresponding revisions changes in the re-submitted files.

2. Point-by-point response to Comments and Suggestions for Authors

Comments 1: [Abstract: The rootstock utilized with different peach cultivars is the most essential criterion for researching abiotic stresses in temperate fruits crops. There is no information on the rootstock/scion combo. The majority of the time, rootstocks control resistance/tolerance. What is the meaning of bore in ‘inclusive of ELI and MDA, that bore significant’?  Fix it.]

Response 1: Thank you for bringing this to our attention. We fully concur with this comment, and as a result, we have revised the abstract by highlighting the updates in red within the article. The information regarding the rootstock/scion combination can be found in the materials section (2.1. Plant Materials Paragraph 1, line 2).

Comments 2: [Introduction:

1: Not flowing with language. Change the words. ‘osmotic regulating substances’.

2: Do not use fancy word like diving. ‘temperatures diving’. Fix it.

3: It is difficult to determine what causes these trees' cold resistance. Is it the result of rootstock or scion? The authors did not discuss any review of literature on factors responsible for peach cold tolerance. There was no evaluation of the research on rootstocks, how chilling harms rootstock/scion, and how chilling affects oxidative stress and antioxidant defense. The subject's introduction is brief, and necessary references for the parameters under consideration are required.]

Response 2: We agree with these comments.

1: The language of the Introduction has been revised to incorporate additional content and references.

2: I agree with you that rootstock is crucial for the cold resistance of peaches. However, even when using the same rootstock in cultivation, there are still significant differences in cold resistance among scions. Therefore, all resources used in this experiment were grafted onto the same long-standing rootstock commonly found in the area - Hong Hua Shan Tao (P. davidiana Franch). We have also included this information in response to your reminder as it was not clearly explained in the previous version of the article.

3: Additional content and references have been incorporated into the introduction.

Comments 3: [Method:

 1: No need to use fancy word like renowned. ‘renowned Peach Genetic Repository’

2: ‘Samples, comprising 28 distinct germplasms’. Germplasm in what form was requested?

3: ‘On January 5th’. What year?

4: Only rootstock was the part of this study. What is the significance of this work when cold hardiness is governed by a rootstock in a rootstocks/scion combination? In fact, rootstock has the ability to alter the physiology and molecular biology of scions. These points must be clarified by the authors.

Physiological and Molecular Aspects of Plant Rootstock-Scion Interactions. https://www.frontiersin.org/articles/10.3389/fpls.2022.852518/full

5: How many replicates are there per scion (the central section of a branch)? Only three replicates per treatments? How many times was the experiment repeated? One or two times with three duplicates of the middle section of the branch per treatment? N =? Provide this information in a clear manner.

6: Why was no review provided in the introductory section for the following parameters? A review will assist the reader in better understanding the consequences of these parameters?

CR (Cortex ratio of branches) = (thickness of cortex/branch radius) × 100%; CLR (Cork layer ratio) = (cork layer thickness/branch radius) × 100%; XR (Xylem ratio) = (xylem thickness/branch radius) × 100%; X/C = xylem thickness/cortex thickness.]

Response 3:

1: The language of method has been modified.

2: We modified the inappropriate language and detailed the form of the resource. ( 2.1. Plant Materials)

3: The section on Plant Materials has been revised to enhance the clarity and accuracy in conveying the information regarding sample collection and methodology. [2.1. Plant Materials].

4: Thanks to your literature, we concur that the influence of rootstock on the aerial part of fruit trees can be pivotal. However, in Gansu Province's Hexi Corridor region, cultivated varieties grafted onto the same rootstock fail to survive winter, while local peach varieties have endured for over 2,000 years. Henceforth, although rootstock plays a significant role, the above-ground portion also exhibits substantial variation in cold resistance. This study assesses the cold resistance of resources based on identical rootstocks.

5: To enhance the clarity and precision of sample collection and method information, we have revised the section on Plant Materials [2.1. Plant Materials].

6: Relevant details have been incorporated into the introduction.

Comments 4: [Results:

1: At 300% resolution, a graph looks like this. Data visualization is tricky. I delegated the decision on the display clarity of these graphs to the editor. Nothing is visible at 100% resolution.

2: ‘For the nine varieties'. One cannot argue that there are nine varieties, one of which is rootstock and the others are scions. 

3: Generalized data were reported with no consideration for the scions and rootstocks employed. It is nearly impossible to visualize any conclusion using graphs. Authors must report results for each scion and rootstock, as well as how data changes as temperature decreases. This image would benefit breeders rather than the general results.

4: Let me ask a question: at what temperature did the authors find the maximum and lowest proline concentrations, and in which scion and rootstock? Such material in the results section will assist readers better understand the findings of the investigation.

5: “Considering specific varieties, "Hong Hua Shan Tao" exhibited an LT50 value below -28, identifying it as exceptionally cold-tolerant. Conversely, the "Kanoiwa" variety recorded an LT50 of -20.55, marking it as the least cold-hardy among the examined peach cultivars”

This is a technical issue. Why should a rootstock (Hong Hua Shan Tao) be compared to a scion (Kanoiwa)? Alternatively, with other scions. Rootstocks must only be compared to other rootstocks. Why so? Because no one investigates rootstock branches. It is the root that is important. However, in scion, branches are important because they bear fruit.

6: ‘boasting correlation coefficients’. Do use fancy word like boasting].

Response 4:

1:The image has been re-edited and the original version has been submitted to the editorial department.

2:We modify the description of ‘varieties’.

3:  The graph was converted into a line graph in order to enhance the clarity of data fluctuations.[figure 2-3]

4:  The descriptions of physiological and biochemical changes have been incorporated [Section 3.1, Paragraph 3, line 4-11.].

5:To encompass a wide range of peach varieties, we selected 8 types (including peach, flat peach, nectarine) along with a rootstock sourced from the Gansu region. While it is not appropriate to compare varieties with rootstocks in our analysis of results, our objective is to identify indicators that can facilitate the comprehensive evaluation of additional peach resources. We have rectified any improper comparisons in the analysis results by removing inappropriate terminology.

6: Inappropriate words were removed

Comments 5: [Discussion:

1: ‘forming an "S"-shaped curve’. I did not see any such graph in the manuscript. Only bar diagrams.

2: For proline, cat, pod, and sod, there is only generalized discussion with no pertinent references. There is no information or discussion about the selected rootstocks and scions. Surprisingly, the authors did not address antioxidant defense factors in the discussion section.

3: Zhan et al.'s observations [33]. Or Zhang? On sugarcane.

Transcriptome Analysis of Genes Involved in Cold Hardiness of Peach Tree (Prunus persica) Shoots during Cold Acclimation and Deacclimation

4: In the discussion section for anatomical research, not a single rootstock or scion was mentioned. Simply a textbook-style explanation.

Response 5:

1: The form of the graph has been modified (Figure.2).

2: As for the indicators mentioned, the contents of the discussion are enriched [Section 4.1, Paragraph 4, line 23-22.]

3: We fixed the error (Zhang et al.'s observations). [Section 4.1, Paragraph 3, line 21.]

  1. The discourse has been enhanced. [Section 4.2, Paragraph 2, line7-8, line 13-15.]

Round 2

Reviewer 1 Report

Comments and Suggestions for Authors

The authors revised their MS carefully and i have no more comments.

Comments on the Quality of English Language

Fine

Reviewer 2 Report

Comments and Suggestions for Authors

The authors have revised the manuscript substantially, and I recommend it for publication. 

Comments on the Quality of English Language

The authors have revised the manuscript substantially, and I recommend it for publication. 

Reviewer 4 Report

Comments and Suggestions for Authors

1: Check scientific names, which must be italicized. 

2: The authors altered the material in response to feedback. 

3: This is crucial work for local peach growers. I applaud the authors for their efforts.

4: Recommend acceptance after thorough grammar, capital and lowercase, and italics checking. 

Comments on the Quality of English Language

Recommend acceptance after thorough grammar, capital and lowercase, and italics checking.